# Ursolic Acid and Solasodine as Potent Anti-Mycobacterial Agents for Combating *Paratuberculosis*: An Anti-Inflammatory and In Silico Analysis

**DOI:** 10.3390/molecules28010274

**Published:** 2022-12-29

**Authors:** Manthena Navabharath, Varsha Srivastava, Saurabh Gupta, Shoor Vir Singh, Sayeed Ahmad

**Affiliations:** 1Department of Biotechnology, Institute of Applied Sciences & Humanities, GLA University, Mathura 281406, Uttar Pradesh, India; 2Bioactive Natural Product Laboratory, Centre of Excellence in Unani Medicine (Pharmacognosy and Pharmacology), School of Pharmaceutical Education and Research, Jamia Hamdard, New Delhi 110062, Delhi, India

**Keywords:** Crohn’s disease, Dephospho-Coenzyme A kinase (DPCK) protein, in silico, *Mycobacterium avium* subspecies *paratuberculosis* (MAP), REMA assay, solasodine, ursolic acid

## Abstract

*Mycobacterium avium* subspecies *paratuberculosis* (MAP) infection in domestic livestock causes persistent diarrhea, weight loss, and death and is also a potential cause of Crohn’s disease (CD) in humans; notably, treatments against MAP are insufficient, costly, and can cause adverse reactions. Hence, plant-derived bioactive constituents have been taken into consideration in this regard. Herein, we present the results of two bioactive constituents (Solasodine and Ursolic acid) that were evaluated for their safety and efficacy against MAP protein (Dephospho-Coenzyme A kinase (DPCK) by utilizing in vitro assays and different tools of in silico biology. The ADME/*t*-test, the drug-likeness property test, pharmacophore modelling, and PASS prediction have proven that both the constituents have better binding capacities than the available antibiotic drugs used to target protein inhibition pathways. Through our observations, it can be inferred that these two phytochemicals can be adequately used to treat *paratuberculosis,* thereby combating inflammatory bowel disorders (IBD) of an autoimmune nature.

## 1. Introduction

*Mycobacterium avium* subspecies *paratuberculosis* is the causative agent of Johne’s disease (JD), a chronic infectious disease of ruminants that is widespread throughout the world and causes significant production losses and human infection (Zoonotic) (MAP). In ruminants, the disease causes debilitation or cachexia. Due to its zoonotic potential, it has received significant attention. IBD, Crohn’s disease, ulcerative colitis, Type 1 diabetes, thyroiditis, and rheumatoid arthritis are examples of autoimmune diseases and co-morbidities that increase susceptibility to MAP infection [1,2,3,4,5,6,7,8,9,10,11]. When MAP infects susceptible people, it often kills MAP bacilli in the gut, thereby causing inflammatory reactions and possibly harmful inflammatory reactions in other organs. In some cases, chronic diseases have been suspected to be associated with the etiology of *Mycobacterium avium* subspecies *paratuberculosis* (MAP) in domestic livestock. This pathogen is endemic across the globe. It is estimated that around 72.0% of US dairy cattle flocks are infected with Johne’s disease. In the last 28 years, there has been an increasing trend in the bio-load of domestic livestock [12], which is found to be highest (16.0–54.7%) in sheep, followed by buffaloes (28.3–48.0%), cows (6.0–39.3%), and goats (9.4–20%) [13]. Global and Indian researchers have revealed extremely high levels of live MAP bio-loads in raw and liquid milk as well as milk products. Using six tests (Microscopy, IS900 PCR, I_FAT, d_ELISA, i_ELISA, LAT, and cultures), bio-loads of 9.0–60.0%, 10.7–42.1%, and 21.4–71.4% were found in bulk-tank milk. Commercially sold milk products have been tested with respect to their bio-loads of MAP using six tests, which revealed bio-loads of 6.0–50.0%, 13.8–44.4%, 4.7–23.8%, 20.0–40.0%, 12.5–87.5%, 30.0–40.0%, and 50.0% in curds, matka, kulfi, butter, cheese, lassi, buttermilk, and ice cream, respectively [14]. There have been reports of MAP bacilli spreading from animals to people through milk and milk products [12].

Due to the absence of a vaccine for MAP, MAP infection in humans is treated either surgically by removing contaminated intestines or medically by employing anti-tuberculosis medications [15], which have each shown limited success due to established resistance [16]. Due to the increase in cases of animal and human infections, the demand for natural products as an alternative therapy for this chronic, incurable disease has increased. This has encouraged researchers to isolate bio-active (marker) compounds from plants with pharmacological properties for fighting MAP infection.

Drug discovery has been fast-paced due to the rapid advancement in drug research and development. The modern in silico research approaches involving computational analysis and virtual screening can reduce both the time and costs required in the research process. These in silico methods help us to predict molecular docking, pose-interaction, and ligand binding sites within the target molecule. The lower the score, the higher the appreciable interaction between the ligand and receptor [17,18]. Natural bioactive components from plants, such as solasodine and ursolic acid, have been proven to exhibit anti-*paratuberculosis* properties. *Ursolic acid* is a penta-cyclic triterpenoid component present in the *Ocimum sanctum*. It is a biologically active phytochemical that has many different ranges of pharmacological activities, such as antifungal, anti-inflammatory, anti-hypertensive, antioxidant, antibacterial, and anti-ulcerogenic properties. *Solasodine* is a glycol-alkaloidal component present in *Solanum xanthocarpum* which has immune-modulatory, anticancer, antiandrogenic, antipyretic, cardiotonic, anti-spermatogenic, and antifungal effects.

In the present study, the inhibitory roles of ursolic acid and solasodine were evaluated against Dephospho-Coenzyme A (dephospho-CoA) kinase (DPCK), and it was reported that DPCK catalyzes the final step in the biosynthesis of coenzyme A (CoA) in *Mycobacterium avium* subspecies *paratuberculosis* through the ATP-dependent phosphorylation of dephospho-CoA. In this study, a commercially available and abundantly used anti-tuberculosis drug, rifampicin, was employed against MAP Dephospho-Coenzyme A (dephospho-CoA) kinase (DPCK) protein. To test the efficacy and potential of two ligands (solasodine and ursolic acid), they were docked against MAP-DPCK protein. Later, multiple experiments were performed to determine the best ligand, which was then compared with a control to test its efficacy with respect to inhibiting *paratuberculosis* infection.

## 2. Results

### 2.1. Effect of the Bioactive Compounds on Membrane Stabilization Assay

Preliminary anti-inflammatory activity was assessed in vitro, and the bioactive compounds were found to exert protective effects. Erythrocytes against heat-induced hemolysis produced a concentration related effect. Hemolysis induced by heat and thus causing hypotonicity showed protective effects (Table 1).

### 2.2. Determination of Anti-MAP Activity with MIC Using REMA

Preliminary Anti-MAP activity was assessed in vitro, and the activity of bioactive compounds was determined using REMA. The percent inhibition of MAP is depicted in Figure 1, and the inhibition activity, which was measured as MIC_50_, is tabulated in Table 1. Ursolic acid and solasodine showed the highest maximum inhibition activity at the lowest concentration. Concerning their effects as bioactive compounds, the MIC_50_ values were 12 µg/mL and 60 µg/mL for ursolic acid and solasodine respectively.

The most bioactive compounds were determined: OS and SX. The anti-MAP (MIC_50_) values were found to be 12 and 60 µg/mL for both bioactive compounds, namely, OS and SX.

### 2.3. Docking Simulation and Ramachandran Plot Analysis

The control ligands were docked with target receptor proteins along with the selected ligand molecule, MAP Dephospho-Coenzyme A kinase (DPCK) (Figure 2). The docking score of the control protein, rifampicin with MAP DPCK protein, was −7.2 Kcal/mol. The docking scores for solasodine and ursolic acid were −9.0 Kcal/mol and −9.8 Kcal/mol, respectively, with MAP DPCK Protein. The highest number of hydrogen bonds was formed by Rifampicin with MAP DPCK protein. Ursolic acid interacted with seven amino acids, namely, Ala 346, Val 382, Lys 259, Val 261, Trp 317, Phe 345, and Phe 389, comprising a significant number of amino acids, while solasodine interacted with four amino acids, namely, Val 245, Ala 346, Val 382, and His 334, within the DPCK binding pocket when docked against Dephospho-Coenzyme A kinase (DPCK) Protein. Rifampicin (control drug) interacted with all the amino acids (Asn 332, Asp 295, Asp 379, Val 382, Phe 345, Val 296, Val 261, Lys 259, Val 382, and Tyr 381) docked against Dephospho-Coenzyme A kinase (DPCK) Protein. Table 2 lists the docking scores, glide ligand efficiencies, and glide energies of the control. Table 2 illustrates the docking score; glide ligand efficiencies; glide energies; number of hydrogen bonds; number of interacting amino acids as well as the types of bonds and distances between them, which describe the best interactions between ligands and receptors; and the amino acids that participated in the interaction.

### 2.4. Drug-Likeness Properties

Two ligands satisfied Lipinski’s rule of five, which consists of the following criteria: donors of hydrogen bonds in total (acceptable range: ≤5),molecular weight (acceptable range: ≤500), the total number of hydrogen bond acceptors (acceptable range: ≤10),molar refractivity (40–130), and lipophilicity (LogP, acceptable range: ≤5) [19].

Ursolic acid had the apical molecular weight of 456.7 g/mol. Solasodine exhibited a molecular weight of 413.64 g/mol, and Rifampicin showed a high molecular weight of 822.94 g/mol. Ursolic acid showed the highest Log Po/w value (7.0), followed by solasodine (5.2) and Rifampicin (−0.05).

Ursolic acid and Solasodine had two hydrogen bond donors and three hydrogen bond acceptors each. The topological polar surface area (TPSA) of ursolic acid was 57.53 Å^2^. However, the lowest TPSA value was produced by solasodine at 41.49 Å^2^. Ursolic acid exhibited the highest molar refractivity of 132.61. Solasodine showed a molar refractivity of 119.75. Ursolic acid revealed the highest (0.85) and Solasodine the lowest (0.55) drug-likeness score. However, only ursolic acid was found to be effective against paratuberculosis. In addition, Solasodine was found to exhibit the only immune-modulatory effect. All the ligands were found to be anti-inflammatory. The drug-likeness properties values are listed in (Table 3).

### 2.5. Absorption, Distribution, Metabolism, Excretion, and Toxicity (ADME/T) Test

The results of the Absorption, Distribution, Metabolism, Excretion, and Toxicity (ADME/T) test are listed in Table 4.

#### 2.5.1. Absorption

Solasodine showed a high content of positive Pgb substrate and high GI (Gastrointestinal Absorption), while Ursolic acid exhibited low GI absorption. However, none of them were both Pgb inhibitors and substrates.

#### 2.5.2. Distribution

Solasodine showed Blood–Brain Barrier (BBB) permeability, where as ursolic acid did not cross the BBB.

#### 2.5.3. Metabolism

None of the isoenzymes of cytochrome P450 (CYP450) were substrates or inhibitors of solasodine or ursolic acid. However, the CYP450 2D6, CYP450 2C19, CYP450 2C9, and CYP3A4 substrates could not be identified since the data server was unavailable.

#### 2.5.4. Excretion

Solasodine and Ursolic acid exhibited the highest half-lives of 1.7 and 0.5 h.

#### 2.5.5. Toxicity

None of the ligands showed positive values for the Ames and Human hepatotoxic tests. However, Solasodine showed a capacity for drug-induced liver injury.

### 2.6. Pharmacophore Modelling

Two ligands were observed to be able to produce pharmacophore hypotheses while inhibiting the MAP Dephospho-Coenzyme A kinase (DPCK) protein during the pharmacophore-modeling experiment. With the MAP Dephospho-Coenzyme A kinase (DPCK) Protein, solasodine began a four-point assumption (Characteristics: B1, B2, E3, and S4) and generated one pi-cation bond, one ugly bond, three nasty bonds, and one hydrogen bond with the protein-binding pocket. While inhibiting the MAP Dephospho-Coenzyme A Kinase (DPCK) Protein, ursolic acid produced four-point assumptions (Characteristics: B1, B2, E3, and S4) and created one hydrogen bond and generated one pi–pi interaction within the receptor-binding pocket. However, both ligands displayed a high number of bonds with their respective receptor proteins (Figure 3).

### 2.7. Prediction of Activity Spectra for Substances (PASS) Prediction Study

In the Prediction of Activity Spectra for Substances (PASS) prediction study, the LD50 value of solasodine was predicted; however, due to the unavailability of information via a database server, toxicity was not determined. However, for ursolic acid, an LD50 value of 300 mg/kg was predicted. The PASS prediction study inferred behaviors for the 35 deliberate biological activities with no observed toxic effects. The PASS prediction experiment was carried out, incorporating a threshold value (Pa > 0.7) yielding a highly decisive prediction [20]. Both solasodine and ursolic acid showed various pharmacological activities. No toxic effects were shown by solasodine and ursolic acid. The results of the Prediction of Activity Spectra for Substances (PASS) Prediction Studies are listed in (Table 5).

## 3. Discussion

In molecular-docking studies, scores are generated by comparing the attachment of ligands to receptors. In 2019, Sarkaret al. showed that bonding affinity decreases with an increase in binding energy, and vice versa [21]. The best results, according to certain studies, are compared to those with the lowest glide energy [22]. Rifampicin successfully docked with its target receptor when used as a control. The control’s docking score was −7.2 Kcal/mol, which is a good result. *Ursolic acid* had the lowest docking score (−9.0 Kcal/mol) and *solasodine* had the greatest docking score (−9.8 Kcal/mol) against the MAP DPCK protein. The binding affinity of solasodine to MAP DPCK was higher than that of ursolic acid.

The drug discovery and development process improves the estimation of drug-likeness properties. A drug’s topological polar surface area (TPSA) and molecular weight affect its permeability with respect to biological barriers. As its molecular weight and topological polar surface area (TPSA) increase, a drug’s permeability decreases, and vice versa. In both organic and aqueous phases, the value of lipophilicity is stated in terms of LogP. The logarithm of the candidate molecule’s barrier coefficient is known as LogP. The lipophilicity of a drug molecule affects its absorption inside the body. The higher the value of LogP, the lower the absorption, and vice versa [23]. The solubility of the candidate molecule is influenced by the lowest value of LogS. The interaction also depends upon the number of hydrogen bonds. The greater the number of hydrogen bonds, the higher the energy, and vice versa [24]. Based on the drug-likeness property experiment, it was found that ursolic acid had a high molecular weight of 456.0 g/mol, a high LogP value (5.93), and a drug likeness score of 0.85, with no tumorigenic effects, irritant properties, or reproductive effects. In addition, it was established that ursolic acid has the highest TPSA score of 57.53 Å^2^ and exhibited mutagenic properties. Solasodine, in comparison with ursolic acid, showed a good LogS value with a molecular weight of 413.0 g/mol and a drug score of 0.55. Solasodine had a TPSA score of 41.49 Å^2^ with no irritant properties or influence on reproductive effectiveness. However, tumorigenic effects were found. The Lipinski rule of five was observed by all of the ligands.

The ADME/T test is used to assess a drug’s pharmacological and pharmaco-dynamic properties inside a biological system. Drugs targeting brain cells and the blood–brain barrier play a vital role in their function. Since the oral delivery of medications is the preferred method, it is assumed that the majority of medications are absorbed in the intestine. P-glycoprotein in cells facilitates the transfer of several medicines. As a result, its inhibition has an impact on the drug delivery mechanism.

Drugs taken orally enter the bloodstream through the stomach and liver. They are broken down by a set of enzymes in the liver and eliminated through urine or bile. A pharmacological factor that affects the pharmaco-dynamics, as well as the circulation and elimination of medications, is the binding of pharmaceuticals to plasma proteins. The efficiency of a drug depends upon how well it binds to plasma proteins [25,26]. A drug can be transmitted without lethality through the cell membrane less efficiently—and vice versa—if it binds to the proteins limited to plasma. The better the half-life of a drug, the longer it stays in the body, which is why the drugs’ half-lives are calculated [27,28]. Many agents can block the HERG channel, resulting in cardiac arrhythmia and even mortality. The metabolism of xenobiotics takes place in the liver, which is where their toxic effects are studied. The Ames test is a mutagenicity test used to determine which substances are mutagenic. Chemicals known to cause mutations are also known to cause cancer. Drug-induced liver damage (DILI) is brought on by drug consumption. It is one of the elements that can result in a number of liver problems. [29,30].

The ligands performed similarly with respect to the absorption section of the experiment, however, based on the probability values, the most outstanding performers of the absorption section were *solasodine* and *ursolic acid*. Both ligands had high plasma protein affinity in the distribution region, and they were all blood–brain barrier-permeable. Based on the probability values, *solasodine* would be the best ligand in the distribution section. No ligand performed well with respect to the metabolism section. However, *ursolic acid* demonstrated relatively good results since it did not inhibit any of the CYP450 isoenzymes. Based on the likelihood estimates, *solasodine* would be the ideal ligand in the metabolic section. Among the excretion ligands, *solasodine* has the highest half-life of 1.8 h. Regarding toxicity, *solasodine* was positive in both the Ames test and Human hepatotoxic test. It also showed the ability to cause drug-induced liver injury.

The phase pharmacophore perception engine of ZINC Pharmer is used to screen 3D databases, model pharmacophores, and develop QSAR models. Modeling pharmacophores is mostly based on the six sorts of built-in features of this engine: hydrogen bond acceptor (A), hydrogen bond donor (D), negatively ionized (N), positively ionized (P), hydrophobic (H), and aromatic ring (R) features. Customization can lead to an increase in the number of qualities. For the chosen ligand molecules, pharmacophore modeling generates hypotheses that can be applied to various in vitro and in vivo studies [31]. All the ligand molecules successfully generated the hypotheses. Based on the experiments conducted, two ligands were selected as the most effective in inhibiting the MAP Dephospho-Coenzyme A kinase (DPCK) Protein. *Solasodine* exhibited the highest docking score, 9.8 kcal/mol, while *Ursolic acid* generated a docking score of 9.0 kcal/mol with respect to inhibiting the MAP Dephospho-Coenzyme A kinase (DPCK) Protein. The drug-likeness property experiment showed that both *solasodine* and *ursolic acid* generated fairly good results in the ADME/T test. The findings of the ADME/T test were better for *solasodine* but worse for *ursolic acid*. Similar to the ADME/T experiment, *solasodine* performed best in the CYP450 SOM experiment, whereas *ursolic acid* performed poorly. On the other hand, *Solasodine*, a drug that inhibits MAP Dephospho-Coenzyme A kinase (DPCK), also showed great results in the pharmacophore mapping and modeling trials. *Ursolic acid*, however, did not show such excellent results (pharmacophore mapping was not determined). As a result, both *solasodine* and *ursolic acid* had high performance and efficiency against ParaTB when compared to the extensively used drugs that are used to treat paraTB.

In this PASS prediction study, only the two best ligands were evaluated in order to ascertain their varied biological and toxicological consequences in a non-pregnant wistar rat (model). According to the five classes of the Globally Harmonized System of Classification and Labeling of Chemicals, the ProTox-II service classifies a chemical compound based on its toxicity (GHS) [17]. According to the GHS, class 1—fatal (ingestion dose is LD50 ≤ 5), class 2—fatal (ingestion dose is 5 < LD50 ≤ 50), class 3—toxic (ingestion dose50 < LD50 ≤ 300), class—harmful (ingestion dose300< LD50 ≤ 2000), and class 5—mildly harmful (ingestion dose2000 < LD50 ≤ 5000), thereby comprising the five classes of toxicity as per UNECE (2005), and the ProToxII server adds one more class called class VI—non-toxic (LD50 > 5000). The predicted LD50 value for *solasodine* was 908 mg/kg. The PASS prediction study determined the behaviors exhibited for the 10 deliberate biological activities and no toxic effects were observed. Consequently, the PASS prediction experiment was carried out. Pa > 0.7 was maintained since this threshold gave a highly decisive prediction [20]. Both *ursolic acid* and *solasodine* showed good biological activities, such as anti-inflammatory, antineoplastic, and phosphatase inhibitor and alkylacetyl glycerol phosphatase hydrolase inhibitory activity. Additionally, *Ursolic acid* also showed caspase 3-stimulatory, CYP2J substrate, antiprotozoal (Leishmania), antiviral (Influenza), testosterone 17 beta-dehydrogenase (NADP+)-inhibitory, insulin-promoting, hepatoprotectant, transcription factor NF kappa B-stimulatory, transcription factor-stimulatory, apoptosis agonist, membrane integrity antagonist, diacylglycerol O-acyltransferase-inhibitory, hypolipemic, oxidoreductase inhibitory, wound-healing, hepatoprotective, antineoplastic, nitric oxide antagonist, caspase 8-stimulatory, antinociceptive, mucomembranous-protective, chemopreventive, phosphatase-inhibitory, antipruritic, acylcarnitine hydrolase-inhibitory, protein phosphatase-inhibitory, and antieczematicandnootropic activity. Where as *Solasodine* showed spasmolytic, papaverin-like, diuretic inhibitory, glyceryl-ether monooxygenase inhibitory, and acylcarnitine hydrolase inhibitory activity. There were no toxic effects shown by solasodine andursolic acid. The PASS prediction study demonstrated the superiority of the single best ligand with the fewest side effects. In addition, in vitro anti-MAP activity using the REMA assay showed that the minimum inhibitory concentration (MIC_50_) of the bioactive compounds required to inhibit the growth of MAP was 12 µg/mL and 60 µg/mL, exhibited by ursolic acid and solasodine (Figure 1 and Table 1). As a result, It can be stated that the chosen agent possesses appropriate performance in the tests when compared to the standard anti-mycobacterial first-line (Rifampicin—150 µg/mL) and second-line drugs (Clarithromycin—100 µg/mL and Ofloxacin—72 µg/mL)used as controls. In vitro anti-inflammatory activity using the heat-induced hemolysis (HIH) assay 250 µg/mL showed that the percentage of inhibitory concentration (IC_50_) of the bioactive compounds required to inhibit inflammation was 51.2% and 89.5%, which was exhibited by ursolic acid and Solasodine (Table 1). As a result, it can be stated that the chosen agent has appropriate performance in the tests when compared to the standard anti-mycobacterial first-line (Rifampicin—32.9%) and second-line drugs (Clarithromycin—49.8% and Ofloxacin—55.7%) used as controls.

## 4. Materials and Methods

The NuncTM MicrowellTM Delta-treated 96-Well Nunclon, flat-bottom microplate was purchased from Thermofisher Scientific, Waltham, MA, USA. Ursolic acid, Solasodine, and Middlebrook 7H9 medium were procured from Sigma-Aldrich, St. Louis, MO, USA, and the standard first-line and second-line tuberculosis drugs (Rifampicin, Clarithromycin and Ofloxacin) were obtained from Sigma-Aldrich, St. Louis, MO, USA. Indian Bison Type strain of mycobacteria was procured from the Biotechnology department, GLA University, Mathura, India. In Discovery studio 2021, the ligand–receptor interactions were visualized using Python Molecular Version 1.5.7. Regarding the interactions, the proteins and ligands were prepared and docked; then, grids were generated, as well as graphical representations of the interactions between those molecules. Pub Chem 2D structure of the ligand was downloaded as SDF (Pub Med (“nih.gov”)) and the protein data bank (RCSB PDB: Homepage) provided the receptor’s structure.

### 4.1. Membrane Stabilization Assay

#### 4.1.1. Erythrocyte Suspension Preparation

A total of 2.0 mL of freshly collected blood from 25-year-old healthy men in K_3_EDTA (tripotassium ethylenediaminetetraacetic acid) tubes and blood was used immediately. The supernatant was removed after centrifuging 1.0 mL of blood in an aliquot for 5 min at 3000 rpm in a 1.5 mL eppendorf tube. Equal volume of sterile isotonic saline solution (0.89% *w*/*v* NaCl) was used to wash cell suspension three times until the supernatant was clear and colorless; then, the packed cell volume (PVC) was measured and reconstituted as cellular component with a concentration of 10% *v*/*v* using phosphate-buffered saline (10 mM, pH 7.4) [32].

#### 4.1.2. Heat-Induced Hemolysis (HIH)

Heat-induced hemolysis (HIH) method, as determined by (Chioma A Anosike et al., 2012), was used, with slight modifications [33]. A total of 50 µL of plant extract was dissolved in 2.95 mL of phosphate-buffered solution (ph. 7.4) and mixed with 50 µL of blood cell suspension. Reaction mixture was incubated at 54 °C for 20 min in a regulated-shaking water bath. The reaction mixture was then centrifuged at 2500 rpm for 3 min, and the hemoglobin content in supernatant was determined using a Spectronic 21D spectro photometer (Milton Roy), Andor USA at a 540 nm wavelength. Rifampicin (250 µg/mL)was used as a standard and the control was phosphate-buffered solution. The percentage of hemolysis inhibition by the plant extract was calculated.
Heamolysis inhibition percentage = (1 − D1/D2) × 100
where

D1 = Absorbance of heated reaction control;

D2 = Absorbance of heated test sample.

### 4.2. Preparation of Mycobacterium avium Subspecies Paratuberculosis Suspension

*Mycobacterium avium* subspecies *paratuberculosis* (MAP) inoculum was prepared from a log phase culture after 90–150 days of incubation. To transfer a clump of colonies, sterile McCarthy tubes with 15 mL of Middlebrook 7H9 broth were used. In order to dissolve the clumps, the suspension was vortexed. Afterward, 2 mL of the remaining supernatant was transferred to sterile, flat-bottomed glass test tubes, in which it was left to sediment for 15 min. The suspension’s turbidity was assessed and set to McFarland 0.5, corresponding to a roughly 1.5 × 10^8^ bacterial suspension per mL. Middlebrook 7H9 broth was then used to dilute the suspension in order to achieve the intended typical inoculum of 2.0 × 10^3^ CFU/mL.

#### Resazurin Micro-Titerassay (REMA)

The REMA plate technique was carried out as previously described [34]. In a 96-well microtiter plate (Thermofisher Scientific), 10-fold serial dilutions of bioactive compound solution using 5% DMSO (1mg/mL) of *Ursolic acid* and *Solasodine* from *Ocimum sanctum* and *Solanum xanthocarpum* plants were carried out. MAP inoculums were prepared using Middle brook 7H9 supplemented broth (OADC, PANTA) (Himedia, PA, USA) with a final volume of 100 µL. A growth control without an antibiotic and a sterility control without an inoculum were included on each plate. Before being re-suspended in a tube with 3 mL of Middlebrook 7H9 medium, the turbidity was adjusted to a McFarland 0.5 standard. Then, Middlebrook 7H9 was used to further dilute this suspension at a ratio of 1: 10. A total of 100µL of this suspension was inoculated onto each well of the plate, which were then sealed and incubated at 37 °C. After 14 days, the plates were incubated for 24 h at 37 °C with 32µL of a 0.04% resazurin solution (Resazurin SRL) added to each well. For similar concentrations, the assay was run three times. Blue to pink color shifts indicated bacterial growth, and an ELISA reader was used to determine MIC_50_ values (i-mark Bio-Rad). The lowest drug concentration, or MIC_50_, prevented a complete color change of the resazurin from blue to pink. At 595 nm, resistance or susceptibility was measured.

MIC = Lowest conc. of antibiotic inhibitory growth + highest conc. that allowed growth of microorganisms/2.

### 4.3. Preparation of Protein

Three-dimensional structures of MAP Dephospho-Coenzyme A kinase (DPCK) Protein in PDB format (PDB ID:6N39) were downloaded from the protein data bank (RCSB PDB: Homepage). The protein preparations were implemented in Python Molecular Version (PMV-1.5.7). A heavy Hydrogen atom was added, and bond orders were assigned. Furthermore, water molecules were removed. Lastly, a force field was used to minimize and optimize the structures [35].

### 4.4. Preparation of Ligand and Receptor Grid Generation

*Solasodine* (PubChem CID: 442985) and *Ursolic acid’s* (PubChem CID: 64945) 2D conformations were obtained from PubChem (PubMed (“nih.gov”)). The conformers of the ligand’s 3D structure were visualized using the 3D structure generator of the PyMol tool software (PyMOL pymol.org). Using Python Molecular Version, structures were assembled (PMV-1.5.7). Grids restricted their active sites to the short, precise regions of the receptor protein where ligands dock. Using the default 2.0 scaling factor of the Van der Waals radius and 0.50 charge cutoff, Glide generates a grid that is then subjected to the force field [36,37]. It is produced around the cubic box’s active site. For the docking test, the adjusted volume of the grid box was 24 × 24 × 24.

### 4.5. Pharmacophore Modeling

ZINC Pharmer software’s “Phase pharmacophore perception engine” was used to model the pharmacophores of two different ligands. In addition to manual pharmacophoremodelling was also performed. The radius-scaling factor was fixed to 1.0 throughout the course, and the radii of receptor atoms were assumed to be van der Waals radii. The thickness of the volume shell was restricted to 6.00 * and the surface areas of receptor atoms within 3.00 A of the ligand surface were disregarded. Molecular pharmacophore models were generated for all the ligand molecules in 2D and 3D.

### 4.6. PASS (Prediction of Activity Spectra for Substances) Prediction Study

PASS predictions were performed only for the two ligands chosen to show the best result with respect to inhibiting their respective receptors against the MAP Dephospho-Coenzyme A Kinase (DPCK) Protein. To conduct PASS prediction, we used PubChem’s canonical SMILES (http://pubchem.ncbi.nlm.nih.gov/, accessed on 18 November 2022) based on PASSWay 2’s drug server (http://www.pharmaexpert.ru/passonline/, accessed on 18 November 2022) [20]. The probability “To be active” (Pa) must be greater than 70% in order to forecast PASS because this yields extremely accurate results [38]. Selected compounds’ biological activity and any possible negative effects were predicted using PASS. The LD50 and Toxicity class were predicted in non-pregnant Wistar Rat (model) using the ProTox-II service (http://tox.charite.de/protox II/, accessed on 18 November 2022) [20].

## 5. Conclusions

In this investigation, two potential phytochemicals’—ursolic acid and solasodine activities against *paratuberculosis* were evaluated. Considering all the parameters, the plant-derived phytochemicals have shown high inhibitory activities against the *Mycobacterium avium* subspecies *paratuberculosis* (MAP) DPCK protein targets. Based on in vitro assays and in silico parameters, such as molecular docking, drug-likeness properties, ADMET/*t*-test, pharmacological property analysis, and the PASS prediction study, ursolic acid was observed to possess the highest inhibitory activity against the MAP DPCK target compared to the commercial drug rifampicin (control). In comparison with ursolic acid, another ligand, solasodine, also exhibited inhibitory activity against the MAP DPCK target. Thus, the synergistic combination of both ursolic acid and solasodine is effective and helpful for treating *paratuberculosis* in animals and humans, as herbal formulations are safe to use for the long term while providing minimum toxicity. Further, in vitro and in vivo experiments should also be carried out to confirm the inhibitory activities of both ursolic acid and solasodine against ParaTB for a better understanding of their disease pathogenesis and therapeutics aspects.

## Figures and Tables

**Figure 1 molecules-28-00274-f001:**
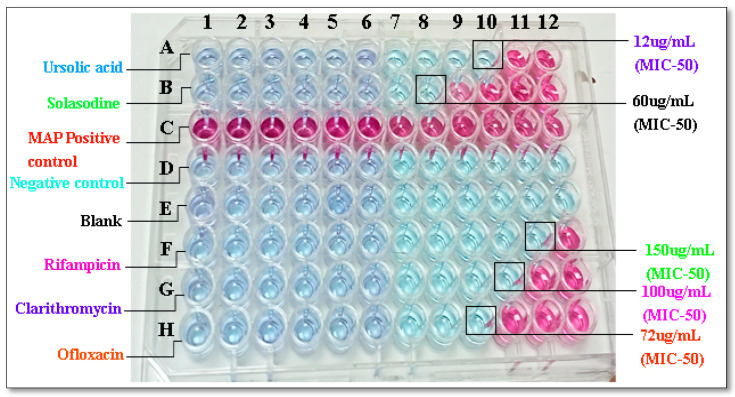
REMA assay. In vitro activity of bioactive compounds, i.e., *Ursolic acid* (12 µg/mL) and *Solasodine* (60 µg/mL), of *Ocimum sanctum* and *Solanum xanthocarpum* and Standard anti-mycobacterial first-and second-line drugs, i.e., Rifampicin (150 µg/mL), Clarithromycin (100 µg/mL), and Ofloxacin (72 µg/mL).

**Figure 2 molecules-28-00274-f002:**
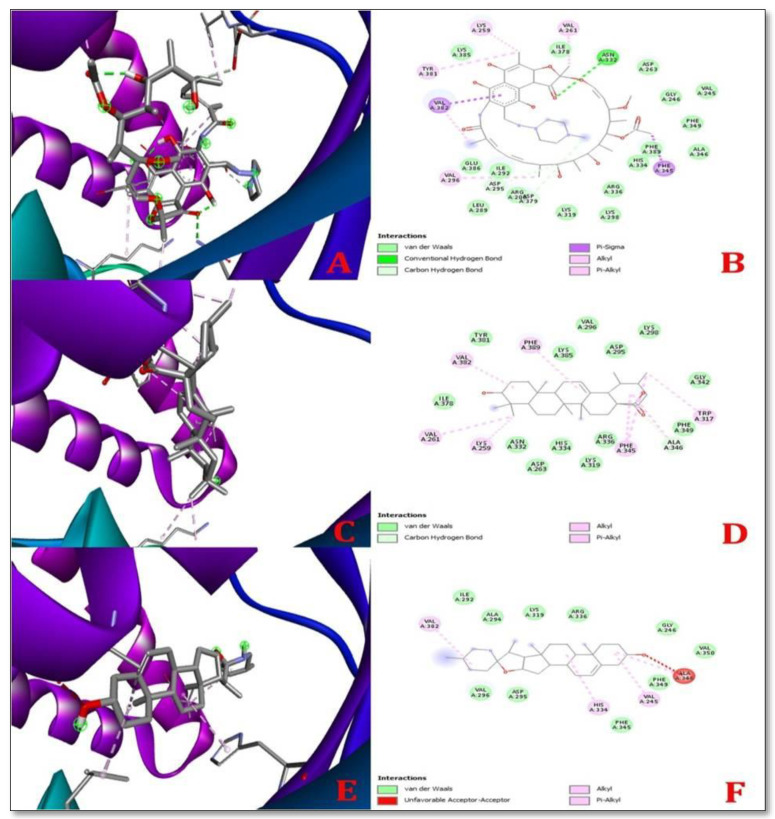
3D (left) and 2D (right) representations of the best pose interactions between receptors and ligandsregarding the MAP Dephospho-Coenzyme A kinase (DPCK) Protein (**A**,**B**) interaction between Rifampicin (control) and DPCK protein, (**C**,**D**). Interaction between ursolic acid and DPCK protein, (**E**,**F**). Interaction between solasodine and DPCK protein. Colored spheres indicate the type of residue in the target: green—conventional hydrogen bond (Asp), light green—carbon–hydrogen bonds (Asp and Ala), pink—pi-sigma (Val and Phe), light pink—alkyl and pi-alkyl (Val, Lys, Tyr, Ala, His, Trp, and Phe), and red—unfavorable acceptor. The protein pocket shows colored lines for the ligands suitable to the nearest atom. The interval between lines illustrates the opening of the protein pocket. In the 3D simulation, the ligands are shown using the stick model and the proteins are represented using the solid ribbon model.

**Figure 3 molecules-28-00274-f003:**
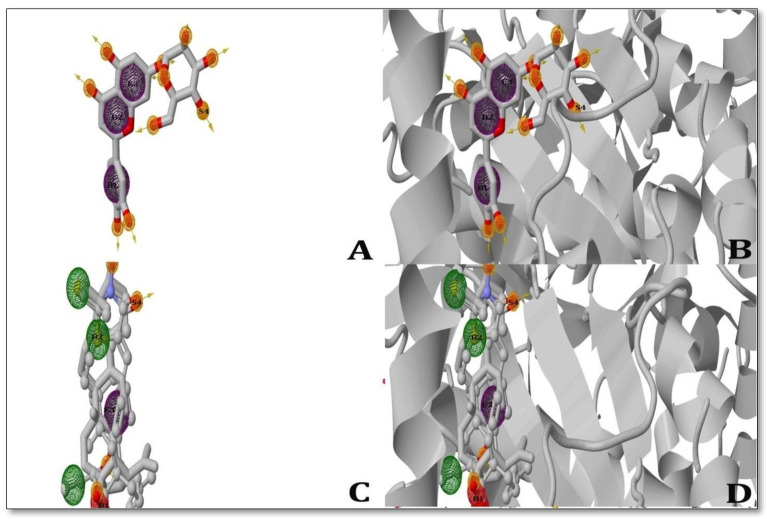
3D depiction of the pharmacophore study used to generate the ligands showing inhibition of the MAP Dephospho-Coenzyme A kinase (DPCK) protein; (**A**) *Solasodine* ligand (left); (**B**) *Solasodine* pharmacophore (right); (**C**) *Ursolic acid* ligand (left), (**D**) *Ursolic acid* pharmacophore (right). The interactions between the ligand and the receptor in the study are shown as ball-and-stick models, where green represents pi-cations; blue represents Alkyl bonds; and yellow represents hydrogen interactions. The poor interaction between the ligands and pharmacophore was analyzed by ZINC Pharmer 2022 version.

**Table 1 molecules-28-00274-t001:** In vitro analysis of bioactive compounds (Ursolic acid and Solasodine) of *Ocimum sanctum* and *Solanum xanthocarpum* using control as a standard and an anti-mycobacterial first-line drug (Rifampicin) and second-line drugs (Clarithromycin and Ofloxacin).

		*Ocimum Sanctum*	*Solanum Xanthocarpum*	Mycobacterial First Line Drug	Mycobacterial Second Line Drugs	
S. No.	Biological Activity	Bioactive Compound (Ursolic Acid)	Bioactive Compound (Solasodine)	Rifampicin	Clarithromycin	Ofloxacin
**1**	IC_50_ value against percentage of heat-induced hemolysis (250 µg/mL)	51.2%	89.5%	32.9%	49.8%	55.7%
**2**	MIC_50_ value against MAP activity mg/mL	12 µg/mL	60 µg/mL	150 µg/mL	100 µg/mL	72 µg/mL

**Table 2 molecules-28-00274-t002:** Molecular docking between *Ursolic acid* and *Solasodine* with receptor in comparison to Rifampicin (control).

S.n	Ligand Name	Receptor Name	Docking Score/Binding Energy (Kcal/mol)	Distance from Best Mode		Interacting Amino Acids	Bond Distance in Å	Types of Bonds
				rmsd 1.b.	rmsdu.b.			
**1**	Control: Rifampicin (PubChem CID: 135398735)	MAP (DPCK) Protein	−7.2	0	0	Asn 332	2.83883	Conventional Hydrogen Bond
						Asp 295	2.64456	Carbon–hydrogen Bond
						Asp 379	3.56655	Carbon–hydrogen Bond
						Val 382	3.26627	Pi-Sigma
						Phe 345	3.93404	Pi-Sigma
						Val 296	5.20374	Alkyl
						Val 261	3.05739	Alkyl
						Lys 259	4.23769	Alkyl
						Val 382	4.82889	Alkyl
						Tyr 381	4.42997	Pi-Alkyl
**2**	*Solasodine* (PubChem CID: 442985)	MAP (DPCK) Protein	−9	0	0	Val 245	5.04793	Alkyl
						Ala 346	4.5958	Alkyl
						Val 382	5.29692	Alkyl
						Hiss 334	4.84765	Pi-Alkyl
**3**	*Ursolic acid* (PubChem CID: 64945)	MAP (DPCK) Protein	−9.8	0	0	Ala 346	3.08779	Carbon–hydrogen Bond
							5.12244	Alkyl
						Val 382	5.14104	Alkyl
						Lys 259	4.31314	Alkyl
						Val 261	5.06138	Alkyl
						Trp 317	4.91095	Pi-Alkyl
						Phe 345	5.01353	Pi-Alkyl
							5.3777	Pi-Alkyl
						Phe 389	4.94191	Pi-Alkyl

**Table 3 molecules-28-00274-t003:** Drug-likeness property studies of selected ligand molecules.

Drug-Likeness Properties	Solasodine	Ursolic Acid
**Lipinski’s rule of five**	Yes	Yes
**Molecular weight (g/mol)**	413.67	456.7
**Concensus Log *Po*/*w***	4.69	5.93
**Log *S***	−4.8	−5.67
**Num. H-bond acceptors**	3	3
**Num. H-bond donors**	2	2
**Ghose**	No	No
**Veber**	Yes	Yes
**Egan**	Yes	No
**Muegge**	No	No
**Molar Refractivity**	127.23	132.61
**TPSA (Å²)**	41.49	57.53
**Druglikeness score**	0.55	0.85

**Table 4 molecules-28-00274-t004:** ADME and toxicity studies of selected ligand molecules.

Class	Properties	Solasodine (with Probability)	Ursolic Acid (with Probability)
**Absorption**	Pgbinhibitor	Negative	Negative
	Pgbsubstrate	Positive	Negative
	GI absorption (Gastrointestinal Absorption)	High	Low
**Distribution**	BBB (Blood–Brain Barrier)	Positive	Negative
**Metabolism**	CYP450 1A2 inhibition	Negative	Negative
	CYP450 3A4 inhibition	Negative	Negative
	CYP450 2C9 inhibition	Negative	Negative
	CYP450 2C19 inhibition	Negative	Negative
	CYP450 2D6 inhibition	Negative	Negative
	Skin permeation	5.00 cm/s	3.87 cm/s
**Excretion**	T1/2 (h)	1.7	0.5
**Toxicity**	DILI (Drug-Induced Liver Injury)	Negative	Negative
	H-HT (Human Hepatotoxicity)	Negative	Negative
	Ames (Ames Mutagenicity)	Negative	Negative
	hERG (hERG Blockers)	Non-blocker	Non-blocker

**Table 5 molecules-28-00274-t005:** Biological activities of selected ligands as per PASS prediction study.

S. No.	Biological Activities	*Solasodine*		*Ursolic Acid*	
		Predicted LD50: NA		Predicted LD50: 300 mg/kg	
		Toxicity Class: NA		Toxicity Class: NA	
		Pa	Pi	Pa	Pi
**1**	Antiinflammatory	0.908	0.004	0.864	0.005
**2**	Spasmolytic, Papaverin-like	0.893	0.003	-	-
**3**	Antineoplastic	0.860	0.006	0.857	0.006
**4**	Diuretic inhibitor	0.828	0.002	-	-
**5**	Glyceryl-ether monooxygenase inhibitor	0.807	0.005	-	-
**6**	Antineoplastic (lung cancer)	0.735	0.005	-	-
**7**	Acylcarnitine hydrolase inhibitor	0.738	0.021	0.748	0.019
**8**	Phosphatase inhibitor	0.717	0.010	0.764	0.005
**9**	Hepatoprotectant	-	-	0.961	0.001
**10**	Transcription factor NF kappa B stimulant	-	-	0.927	0.001
**11**	Transcription factor stimulant	-	-	0.927	0.001
**12**	Antiprotozoal (Leishmania)	-	-	0.915	0.003
**13**	Caspase 3 stimulant	-	-	0.912	0.003
**14**	Apoptosis agonist	-	-	0.890	0.004
**15**	Membrane integrity antagonist	-	-	0.885	0.003
**16**	Diacylglycerol O-acyltransferase inhibitor	-	-	0.882	0.001
**17**	Hypolipemic	-	-	0.885	0.004
**18**	Oxidoreductase inhibitor	-	-	0.876	0.003
**19**	Wound-healing agent	-	-	0.868	0.003
**20**	Antiulcerative	-	-	0.861	0.003
**21**	Hepatic disorders treatment	-	-	0.856	0.003
**22**	Testosterone 17beta-dehydrogenase (NADP+) inhibitor	-	-	0.863	0.012
**23**	Nitric oxide antagonist	-	-	0.843	0.002
**24**	Alkenylglycerophosphocholine hydrolase inhibitor	-	-	0.846	0.012
**25**	Caspase 8 stimulant	-	-	0.834	0.001
**26**	Antinociceptive	-	-	0.821	0.001
**27**	Mucomembranous protector	-	-	0.804	0.017
**28**	Chemopreventive	-	-	0.790	0.004
**29**	Antiviral (Influenza)	-	-	0.761	0.004
**30**	Antipruritic	-	-	0.748	0.005
**31**	Protein phosphatase inhibitor	-	-	0.715	0.003
**32**	Alkylacetylglycerophosphatase inhibitor	-	-	0.717	0.018
**33**	Antieczematic	-	-	0.727	0.037
**34**	Nootropic	-	-	0.708	0.039
**35**	CYP2J substrate	-	-	0.713	0.046

## Data Availability

Data is contained within the article.

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
