# Peer review of "Ursolic Acid and Solasodine as Potent Anti-Mycobacterial Agents for Combating Paratuberculosis: An Anti-Inflammatory and In Silico Analysis"

_molecules, 2022, doi:10.3390/molecules28010274_

Round 1
Reviewer 1 Report
One of the targets for the development of antimicrobial and antiparasitic drugs is the coenzyme A (CoA) biosynthetic pathway. In both Prokaryota and Eucaryota, CoA is an essential cofactor as an acyl group carrier in cellular energy metabolism. The final enzyme in the CoA synthesis pathway is dephospho-CoA kinase (DPCK, EC 2.7.1.24). One of the goals of the research in this paper should be to prove specific, selective binding and inhibition of DPCK, but not its human counterpart. 1. Initially, in pilot studies, this can be demonstrated by in silico modeling, as the Authors did and showed that their research is promising. But I consider it necessary to perform in vitro cytotoxicity tests (at least against normal) Eucaryota cells, which can also be the target of the tested active compounds. PASS – these are prediction activity tests, their results should be confirmed in simple in vitro determination. Then the Therapeutic Index can be calculated, which proves the safety and effectiveness of the new compound. 2. I also propose to prepare and attach the results (Lineweaver-Burk charts) confirming the type of inhibitory activity of Solasodine and Ursolic acid plant compounds against the DPCK enzyme [e.g. like here: Nurkanto A, Imamura R, Rahmawati Y, Prabandari EE, Waluyo D, Annoura T, Yamamoto K, Sekijima M, Nishimura Y, Okabe T, Shiba T, Shibata N, Kojima H, Duffy J, Nozaki T. Dephospho-Coenzyme A Kinase Is an Exploitable Drug Target against Plasmodium falciparum: Identification of Selective Inhibitors by High-Throughput Screening of a Large Chemical Compound Library. Antimicrob Agents Chemother. 2022 Nov 15;66(11):e0042022. doi: 10.1128/aac.00420-22. Epub 2022 Oct 31. PMID: 36314787; PMCID: PMC9664868].
3. Therefore, in the absence of at least basic cytotoxicity tests (MTT), I do not agree with the Authors' declaration in Abstrakt and the Conclusions that the data from the presented in silico modeling tests allow the use of plant compounds in the treatment of paratuberculosis: 'have proved that both the constituents have better binding capacity than the available antibiotic drugs targeting protein inhibition pathway (...)these two phytochemicals can be adequately used to treat paratuberculosis thereby combating inflammatory bowel disorders.' (19-23)
4. Please correct minor grammatical errors and proper names of plants, as well as Latin expressions, write italica, e.g.:
‘ocimum sanctum’ 64
It should be Ocimum sanctum.
‘solanum xanthocarpum’68
It should be: Solanum
‘in vitro’ 83
It should be: in vitro
Author Response
Title: Ursolic acid and Solasodine as Potent Anti-mycobacterial Agents for Combating Paratuberculosis: An Anti-inflammatory and In-Silico Analysis
Manuscript No.: 2076714
Reviewers |
Comments |
Response |
Reviewer 1 |
1. Initially, in pilot studies, this can be demonstrated by in silico modeling, as the Authors did and showed that their research is promising. But I consider it necessary to perform in vitro cytotoxicity tests (at least against normal) Eucaryota cells, which can also be the target of the tested active compounds. PASS – these are prediction activity tests, their results should be confirmed in simple in vitro determination. Then the Therapeutic Index can be calculated, which proves the safety and effectiveness of the new compound. |
Agreed with the suggestions. Active metabolites (Solasodine and Ursolic acid) has also been tested using REMA assay for to confirm the in-vitro toxicity in cells. Definitely we will perform the PASS analysis in future to calculate the therapeutic index of the above metabolites. |
2. I also propose to prepare and attach the results (Lineweaver-Burk charts) confirming the type of inhibitory activity of Solasodine and Ursolic acid plant compounds against the DPCK enzyme [e.g. like here: Nurkanto A, Imamura R, Rahmawati Y, Prabandari EE, Waluyo D, Annoura T, Yamamoto K, Sekijima M, Nishimura Y, Okabe T, Shiba T, Shibata N, Kojima H, Duffy J, Nozaki T. Dephospho-Coenzyme A Kinase Is an Exploitable Drug Target against Plasmodium falciparum: Identification of Selective Inhibitors by High-Throughput Screening of a Large Chemical Compound Library. Antimicrob Agents Chemother. 2022 Nov 15;66(11):e0042022. doi: 10.1128/aac.00420-22. Epub 2022 Oct 31. PMID: 36314787; PMCID: PMC9664868]. |
Agreed and revised as per suggestions. Initially, paratuberculosis is a chronic disease having no botanicals yet identified to treat Mycobacterium avium paraTB completely. So, this in-silico analysis is just the pilot study to know the active pathway involved or can be targeted to inhibit the disease progression which is self challenging using K-10 cattle type strain and compared with our India Bison type biotype, highly prevalent (98.3%) in domestic livestock. Obliviously we have started the in-vivo and in-vitro study to come on final conclusion.
Nurkanto et al., 2022 did the Plasmodium falciparum: Identification of Selective Inhibitors by High-Throughput Screening Dephospho-Coenzyme A Kinase. But we don’t get the selective inhibitors of K-10 MAP strain. |
|
3. Therefore, in the absence of at least basic cytotoxicity tests (MTT), I do not agree with the Authors' declaration in Abstract and the Conclusions that the data from the presented in silico modeling tests allow the use of plant compounds in the treatment of paratuberculosis: 'have proved that both the constituents have better binding capacity than the available antibiotic drugs targeting protein inhibition pathway (...)these two phytochemicals can be adequately used to treat paratuberculosis thereby combating inflammatory bowel disorders.' (19-23)
|
Agreed with the suggestions. Active metabolites (Solasodine and Ursolic acid) has also been tested in-vitro using REMA assay (same as alternative to MTT assay) for confirmation of MIC-50 values for above compounds as pilot study (Phase I). Conclusions part have been as this study is only showing the data from in-silico modelling, on the basis of that we just have develop an idea to get maximum success to treat paratuberculosis (few studies are available on this MAP DPCK pathways inhibition). However, further confirmation can be done by in-vitro and in-vivo studies to evaluate safety and efficacy (phase 2 started).
|
|
4. Please correct minor grammatical errors and proper names of plants, as well as Latin expressions, write italica, e.g.: ‘ocimum sanctum’ 64. It should be Ocimum sanctum. ‘solanum xanthocarpum’68 It should be: Solanum ‘in vitro’ 83 It should be: in vitro |
Agreed and thanks for pointing out the critical point. Done as per your suggestions. |
Reviewer 2 Report
The in vitro and in silico investigation of Solasodine and Ursolic acid have been carried out .
There are grammatical errors and typos. The names of the subsections should be presented in more detail. For example, section 4.3: “Preparation of protein” – what protein, for what? ABB section is needed.
The countries-manufacturers are not indicated, the methods are not described clearly enough. Terms such as "great results", "good", "bad", "great", "excellent" should be replaced by specific values and the reliability of the differences.
Section “Materials and Methods” does not contain any information about which solutions were obtained and how they were obtained. It is written “Plant extract” and nothing more.
Contradictions and ambiguities are caused by the program that was used to predict toxicity and pharmacological activity. For example, insulin promoter, transcription factor NF kappa B stimulant, transcription factor stimulant, apoptosis agonist – there are no such activities in pharmacology.
Some comments:
1. Is Fig.1 a photo? it is desirable to give a scheme, a photograph of the 96-well plate is the primary data, so it is not customary to present the results.
IC50 in table 1 – the units are not indicated. What is “Solanum xanthocarpum Schrad. &Wendl.”?
2. Line 331: “Freshly collected 2.0 ml of blood from healthy humans into K3 EDTA tubes and stored in 4oC for 24 hours before use”.
It is not written – men or women, age, whole blood? What is K3? The storage could influence the blood properties. Why the blood was not used immediately?
3. Line 352: “(MAP) inoculum was prepared from a log phase culture after 90–150 days of incubation.”
What Is the concrete incubation time interval, why it is so long? Why there are no any explanations?
4. Line 359: “ 2.0 10 3 CFU/mL.”
What is the correct concentration? CFU, MIC-50, should be placed into ABB section. Also the combination of the plants “Ocimum sanctum & Solanum xanthocarpum” not clear. What was the exact biomass for the solution, how the solution was prepared : all it should be scrupulously described.
5. It is not clear how the MIC-50 was calculated, what kind of Software was used.
6. “Rifampicin used as a standard and control as a phosphate buffered solution”
There is no concrete concentration of Rifampicin
7. Line 195: “In the Prediction of Activity Spectra for Substances (PASS) prediction study, LD50 value 195 of solasodine was predicted but due to the non-availability of information in a database server, toxicity was not determined. However, for ursolic acid, an LD50 value of 300 mg/kg was predicted”.
Such an explanation is unacceptable in a scientific chemical journal. Then it is needed to choose another program. In addition, from the point of view of molecular and clinical pharmacology, it is impossible to predict LD50. Please, explain, what kind of data and parameters does this program use to predict pharmacological activities and LD50? What is the procedure of predicting? Does the program take into account the routes of administration of drugs - intravenously, orally, etc.? The predicted LD50 for what kind of animals?
8. In the section “Discussion” there is no comparative analysis of the obtained IC50 and MIC-50.
I found in the article MIC for OFX, for example, and it was 0,25 (mg/L ) . Please, see for this [Neetu Kumra Taneja, Jaya Sivaswami Tyagi, Resazurin reduction assays for screening of anti-tubercular compounds against dormant and actively growing Mycobacterium tuberculosis, Mycobacterium bovis BCG and Mycobacterium smegmatis, Journal of Antimicrobial Chemotherapy, Volume 60, Issue 2, August 2007, Pages 288–293, https://doi.org/10.1093/jac/dkm207].
9. Line 410: “The LD50 and Toxicity class were predicted using the Pro- 410 Tox-II service (http://tox.charite.de/protox II/) [20].”
I did not find any results from this service in the paper.
Author Response
Title: Ursolic acid and Solasodine as Potent Anti-mycobacterial Agents for Combating Paratuberculosis: An Anti-inflammatory and In-Silico Analysis
Manuscript No.: 2076714
Reviewers |
Comments |
Response |
Reviewer 2 |
1. Is Fig.1 a photo? it is desirable to give a scheme, a photograph of the 96-well plate is the primary data, so it is not customary to present the results.
IC50 in table 1 – the units are not indicated. What is “Solanum xanthocarpum Schrad. &Wendl.”?
|
Agreed, Thank you for valuable comment. As per your suggestions, the said changes have been incorporated in the manuscript. Fig.1 a photo is added to a results and also I have revised Solanum xanthocarpum & Table no.1 mentioned with IC50 units. |
2. Line 331: “Freshly collected 2.0 ml of blood from healthy humans into K3 EDTA tubes and stored in 4oC for 24 hours before use”.
It is not written – men or women, age, whole blood? What is K3? The storage could influence the blood properties. Why the blood was not used immediately? |
Thanks for pointing out the critical point. It has been corrected as per your suggestion .I have corrected K3EDTA, men and his age already I have mentioned 2.0 ml blood not in whole blood and also The storage could influence the blood properties- Yes, Cold storage of RBC in blood has been reported to be associated with changes in various RBC properties, including increased cell volume. |
|
3. Line 352: “(MAP) inoculum was prepared from a log phase culture after 90–150 days of incubation.”
What Is the concrete incubation time interval, why it is so long? Why there are no any explanations? |
Agreed, Thank you for your keen observation. Yes M. avium subsp. Paratuberculosis(MAP) is a very extreme slow-growing bacilli in middlebroken 7H9 liquid media it will take to come log phase minimum 90-120 days is required. |
|
4. Line 359: “ 2.0 10 3 CFU/mL.”
What is the correct concentration? CFU, MIC-50, should be placed into ABB section. Also the combination of the plants “Ocimum sanctum & Solanum xanthocarpum” not clear. What was the exact biomass for the solution, how the solution was prepared : all it should be scrupulously described. |
Agreed, Thank you for your suggestion. It has been corrected as per your suggestion. Line 359: 2.0 X103 CFU/mL. Bioactive compounds of Ursolic acid & Solasodine from Ocimum sanctum and Solanum xanthocarpum plants and The exact biomass for the solution friable biomasses. The Bioactive compounds solution was prepared using 5% DMSO.
|
|
5. It is not clear how the MIC-50 was calculated, what kind of Software was used. |
Agreed, Thank you for your keen observation. I have been used a formula MIC= Lowest conc. of antibiotic inhibitory growth + highest conc. allow growth of microrganisms / 2 |
|
6. “Rifampicin used as a standard and control as a phosphate buffered solution”
There is no concrete concentration of Rifampicin.
|
Thanks for pointing out the critical point. It has been corrected as per your suggestion. Rifampicin concentration is 250µg/mL used as a standard by Ref. (Chioma A Anosike et al., 2012) |
|
7. Line 195: “In the Prediction of Activity Spectra for Substances (PASS) prediction study, LD50 value of solasodine was predicted but due to the non-availability of information in a database server, toxicity was not determined. However, for ursolic acid, an LD50 value of 300 mg/kg was predicted”.
Such an explanation is unacceptable in a scientific chemical journal. Then it is needed to choose another program. In addition, from the point of view of molecular and clinical pharmacology, it is impossible to predict LD50. Please, explain, what kind of data and parameters does this program use to predict pharmacological activities and LD50? What is the procedure of predicting? Does the program take into account the routes of administration of drugs - intravenously, orally, etc.? The predicted LD50 for what kind of animals? |
Agreed with your suggestions, but due to non-availability of toxicity data on available servers for the PASS study, as only few studies are present on solasodine and their data explored. However, lot of work has been done on Ocimum palnt (Ursolic acid), so maximum data and their biological activity is available to calculate the LD50 of Ursolic acid (active compound). Accordingly, Data only on biological activities of Solasodine are available so which we have mentioned in Table No. 5.
By choosing another program, same problem will occur due to non-availability of toxicity information of Solasodine in database server.
ProTox-II webserver is specific to calculate the metabolite biological activities, LD50 and toxicity information after oral route of administration.
Predicted LD50 was calculated using non-pregnant Wistar Rat (model).
|
|
8. In the section “Discussion” there is no comparative analysis of the obtained IC50 and MIC-50.
I found in the article MIC for OFX, for example, and it was 0.25 (mg/L). Please, see for this [Neetu Kumra Taneja, Jaya Sivaswami Tyagi, Resazurin reduction assays for screening of anti-tubercular compounds against dormant and actively growing Mycobacterium tuberculosis, Mycobacterium bovis BCG and Mycobacterium smegmatis, Journal of Antimicrobial Chemotherapy, Volume 60, Issue 2, August 2007, Pages 288–293, https://doi.org/10.1093/jac/dkm207].
|
Thank you for your keen observation. The discussion of comparative analysis of the obtained IC-50 and MIC-50 has been modified & shown in Table no 1 as per your suggestions. |
|
9. Line 410: “The LD50 and Toxicity class were predicted using the Pro- 410 Tox-II service (http://tox.charite.de/protox II/) [20].”
I did not find any results from this service in the paper. |
Agreed, Thanks for your observation. ProTox-II online webserver was used for toxicological assessment (LD50, etc.) along with molecular mechanisms of toxicity. Detailed results have been shown in Table No. 5. We can also get raw data about the toxicity be accessed using pro-410 Tox-II web sever but sometimes server get temporary down/ slow. |
Round 2
Reviewer 1 Report
The authors have proven antimicrobial and anti-inflammatory activity. To talk about a potential new drug for animals, you need to show cytotoxicity tests and the value of the Therapeutic Index TI. The authors still claim that this in-silico analysis is just the pilot study to know the active pathway involved or can be targeted to inhibit the disease progression. Therefore, I still recommend attaching the TI result (after performing the basic MTT test) or changing the title and abstract to emphasize that this is only a pilot study, and not, as the authors claim, molecules to fight paratuberculosis were obtained.
Author Response
Title: Ursolic acid and Solasodine as Potent Anti-mycobacterial Agents for Combating Paratuberculosis: An Anti-inflammatory and In-Silico Analysis
Manuscript No.: 2076714
Reviewer |
Comments |
Response |
Reviewer 1 |
The authors have proven antimicrobial and anti-inflammatory activity. To talk about a potential new drug for animals, you need to show cytotoxicity tests and the value of the Therapeutic Index TI. The authors still claim that this in-silico analysis is just the pilot study to know the active pathway involved or can be targeted to inhibit the disease progression. Therefore, I still recommend attaching the TI result (after performing the basic MTT test) or changing the title and abstract to emphasize that this is only a pilot study, and not, as the authors claim, molecules to fight paratuberculosis were obtained. |
Agreed with the suggestions. Active metabolites (Salsodine and Ursolic acid) have been tested using REMA assay for to confirm the in-vitro anti-MAP activity. Kindly please refer the reference (G. P. S. Jadaunet,al,2007(https://doi.org/10.1093/jac/dkm117) and also we did the Anti-inflammatory activity Kindly please refer the reference (Chioma A Anosike et al., 2012).Definitely we will perform in future to calculate the therapeutic index of the above metabolites. MTT assay for cytotoxicity test of cell lines this is invitro cell lines studies. |

Reviewer 2 Report
Dear Authors,
Not all the comments have been taken into account and some answers were addressed to the reviewer, but finally have not been added to the text of the manuscript.
Fig.1 is a photo, it looks strange for the article, it is my own opinion.
There are some points below:
Point 2. Why the blood was not used immediately? There is no explanation in the manuscript: is it important for the study or not. If not, then it should be mentioned.
Point 4. The authors answer: ” The Bioactive compounds solution was prepared using 5% DMSO”. But there is no any mention about DMSO in the text of the manuscript.
Also, the final concentration of DMSO should be written because it can significantly influence the results. It should be noted, have DMSO any influence on the results or not, as DMSO is pharmacologically active compound with anti-inflammatory and antimicrobial action.
Point 7. The authors answer: “Predicted LD50 was calculated using non-pregnant Wistar Rat (model)”.
There is no any mention about the model in the text of the manuscript. It should be added to the text.
What was the procedure to predict LD50 and what Pharmacological activities? Please, could you explain it in the text of the manuscript? What big data was used for it and how the predicted activities correlate with the real ones, can you introduce some examples. Is there any convincing evidence? For the ursolic acid there are many articles (for example, Seo DY, Lee SR, Heo JW, No MH, Rhee BD, Ko KS, Kwak HB, Han J. Ursolic acid in health and disease. Korean J Physiol Pharmacol. 2018 May;22(3):235-248. doi: 10.4196/kjpp.2018.22.3.235. Epub 2018 Apr 25. PMID: 29719446; PMCID: PMC5928337 and others). For solasodin:
Malik A, Arooj M, Butt TT, Zahid S, Zahid F, Jafar TH, Waquar S, Gan SH, Ahmad S, Mirza MU. In silico and in vivo characterization of cabralealactone, solasodin and salvadorin in a rat model: potential anti-inflammatory agents. Drug Des Devel Ther. 2018 May 24;12:1431-1443. doi: 10.2147/DDDT.S154169. PMID: 29872266; PMCID: PMC5973396 and others.
Author Response
Title: Ursolic acid and Solasodine as Potent Anti-mycobacterial Agents for Combating Paratuberculosis: An Anti-inflammatory and In-Silico Analysis
Manuscript No.: 2076714
Reviewer |
Comments |
Response |
Reviewer 2 |
Fig.1 is a photo, it looks strange for the article, it is my own opinion. |
Agreed, with the suggestions. It is not look strange for the article. Kindly please refer the reference (Giftania Wardani et, al 2018) this journal authors also representing the figure same like that. |
Point 2. Why the blood was not used immediately? There is no explanation in the manuscript: is it important for the study or not. If not, then it should be mentioned. |
Agreed, Thank you for your keen observation. The Why the blood was not used immediately? has been modified as per your suggestion. |
|
Point 4. The authors answer: ” The Bioactive compounds solution was prepared using 5% DMSO”. But there is no any mention about DMSO in the text of the manuscript. Also, the final concentration of DMSO should be written because it can significantly influence the results. It should be noted, have DMSO any influence on the results or not, as DMSO is pharmacologically active compound with anti-inflammatory and antimicrobial action. |
Agreed, Thank you for your keen observation. The DMSO concentration has been modified as per your suggestion. |
|
Point 7. The authors answer: “Predicted LD50 was calculated using non-pregnant Wistar Rat (model)”.
There is no any mention about the model in the text of the manuscript. It should be added to the text. |
Agreed, Thank you for your keen observation. The non-pregnant Wistar Rat (model) has been modified as per your suggestion. |
|
What was the procedure to predict LD50 and what Pharmacological activities? Please, could you explain it in the text of the manuscript? What big data was used for it and how the predicted activities correlate with the real ones, can you introduce some examples. Is there any convincing evidence? For the ursolic acid there are many articles (for example, Seo DY, Lee SR, Heo JW, No MH, Rhee BD, Ko KS, Kwak HB, Han J. Ursolic acid in health and disease. Korean J Physiol Pharmacol. 2018 May;22(3):235-248. doi: 10.4196/kjpp.2018.22.3.235. Epub 2018 Apr 25. PMID: 29719446; PMCID: PMC5928337 and others). For solasodin: Malik A, Arooj M, Butt TT, Zahid S, Zahid F, Jafar TH, Waquar S, Gan SH, Ahmad S, Mirza MU. In silico and in vivo characterization of cabralealactone, solasodin and salvadorin in a rat model: potential anti-inflammatory agents. Drug Des Devel Ther. 2018 May 24;12:1431-1443. doi: 10.2147/DDDT.S154169. PMID: 29872266; PMCID: PMC5973396 and others. |
Agreed with your suggestions, The briefly explained in Discussion part kindly refer Discussion of manuscript and Table No.5 ProTox-II webserver is specific to automatically calculate the metabolite biological activities, LD50 and Pharmacological activities, toxicity information. Predicted LD50. Data only on biological activities of Ursolic acid & Solasodine are available so which we have mentioned in Table No. 5.
|
